SOFTWARE

# MobsPy: A programming language for biochemical reaction networks

**Fabricio Cravo**[1,2,3], **Gayathri Prakash**[1,4], **Matthias Függer** [1*], **Thomas Nowak** [1,5*]

**1** Université Paris-Saclay, CNRS, ENS Paris-Saclay, LMF, Gif-sur-Yvette, France, **2** Université Paris-Saclay, CNRS, LISN, Gif-sur-Yvette, France, **3** Northeastern University, Boston, Massachusetts, United States of America, **4** Rice University, Houston, Texas, United States of America, **5** Institut Universitaire de France, Paris, France

\* mfuegger@lmf.cnrs.fr (MF); thomas@thomasnowak.net (TN)

## Abstract

Biochemical Reaction Networks (BCRNs) model species and their interactions via reactions. They have been extensively used in chemistry and extended to biological settings by generalizing the reactions' kinetics. However, detailed models of biochemical processes tend to result in complex BCRN models. We present the Meta-species Oriented Biosystem Syntax (MobsPy), a language designed to simplify the modeling process using the concept of meta-species. Meta-species are constructed using a bottom-up approach from base species, which represent elementary, simple characteristics. These characteristics are then combined to create meta-species with all their complex behavior. The combined species have characteristics that are the Cartesian product of the base species' characteristics and feature inheritance of reactions involving the base species. New reactions can involve all the states of a meta-species or only a subset that is selected via a query. In particular, reactions of meta-species can express a state change of one of the reactants. MobsPy is deployed as a Python package. We showcase its modeling capabilities by building concise models for biochemical systems from the literature.

**Data availability statement:** The code for this submission is available online at: https://github.com/ROBACON/mobspy.

## Author summary

The formalism of Biochemical Reaction Networks is a way to directly express a system by its species and the reactions that act upon them. Despite their natural correspondence to modeling biochemical systems, such networks may be difficult to write and maintain: complex networks have hundreds of species and reactions. We provide a new Python-based language, MobsPy, designed to simplify this process by using two concepts: First, one uses meta-reactions to specify patterns that are common to multiple reactions. Second, meta-species can be used to specify the behavior of several species at the same time. Based on these concepts the language provides a means to create species and reactions

**Funding:** This work was supported by the French National Research Agency (ANR, https://anr.fr/en/) under grant numbers ANR-21-CE48-0003 and ANR-23-CE45-0013. (M.F., T.N.) The funding body did not play any role in the study design, data collection and analysis, decision to publish, or preparation of the manuscript.

**Competing interests:** The authors have declared that no competing interests exist.

via inheritance from other species. We demonstrate the capabilities of MobsPy at the hand of several examples from the literature and compare them to existing tools.

## Introduction

A Chemical Reaction Network [1] (CRN) comprises a set of species and reactions among the species. An example is the reaction $A + B \rightarrow C$, where $A$ and $B$ form a complex $C$ upon interaction. While CRNs assume reaction rates to follow mass-action kinetics, i.e., proportionality to concentrations $[A]$ and $[B]$ in the example, the more general Biochemical Reaction Networks [2] (BCRNs) allow for more general rates as they may appear when involving complex species like bacteria.

BCRNs are a powerful formalism used for modeling biochemical systems, with the possibility of separating species interaction from reaction kinetics and immediately translating to both deterministic and stochastic dynamics. Stochastic simulations of BCRNs allow the use of algorithms like the Gillespie algorithm [3], which efficiently handles stochastic trajectories of species. BCRNs have been successful in modeling the experimental behavior of several biochemical systems, with examples being gene expression [4], phage transmission [5], controllers of gene expression [6], and metabolic engineering [7]

However, writing BCRNs can become a challenging endeavor as the complexity of the model grows [8], resulting in numerous reactions and species [8,9]. It is not uncommon for models to reach more than 50 reactions [10,11], with even a single genetic XOR gate reaching up to 62 reactions with 44 species [11]. As models expand, their complexity can lead to errors during the modeling process and in subsequent model updates [8]. Lopez, Muhlich, Bachman, and Sorger [8] examined existing literature and found discrepancies between descriptions of models and the actual provided models. In response to the challenges posed by increasingly intricate BCRN models, researchers have pursued the development of numerous specialized tools and languages aimed at modeling CRNs and BCRNs [8,9,12–20]. Not all of these modeling tools prioritize reaction syntax, with some focusing primarily on simulation. We briefly sketch some of these tools as examples: COPASI [15] is a GUI-based framework written in C++ that provides deterministic and stochastic simulations of CRNs among several analysis and parametrization methods. BasiCO [12] is a Python API that allows COPASI to be used directly in Python. Antimony [18] is a language that allows one to specify models in succinct human-readable code with language features that ease the creation of shareable modules. Related is the simulation framework Tellurium [19] that allows specifying models in Python code, including entries in Antimony, and provides extensive stochastic and deterministic simulation, analysis, and parametrization support. As a simulation backend, it uses libRoadrunner [20]. Both COPASI and Tellurium provide model export to the standardized SBML format [16] with the possibility to run simulations in different backends. iBioSim [17] is a GUI-based BCRN simulation tool that also includes features such as compartments and model generation from genetic design entries. Common to the above tools, however, is that they lack means for reaction modularity. Several tools featuring reaction modularity already exist in the literature, including rule-based languages such as Kappa [13], BioNetGen [21], PySB [8], and STOCHSIM [22]. Additionally, dedicated tools support the implementation of these languages, such as SmolDyn [23] and VCell [24], both of which integrate BioNetGen. BioCRNpyler provides abstractions in terms of mechanisms, components, and mixtures.

Here we introduce MobsPy (Meta-Species Oriented Biosystem Syntax in Python), which is a new Python-based programming language aimed at decreasing the complexity of BCRN model entry, simulation, and analysis.

For model entry and analysis, MobsPy uses the concepts of meta-species and meta-reactions. A meta-species is a set of species, with a species representing the state of the meta-species, i.e., a particular assignment of characteristics and meta-reactions are reactions between meta-species. Meta-species can be directly defined or constructed via the Cartesian product of predecessor meta-species. By forming the product, an orthogonal state space is created, with each predecessor species representing a base in this state space. Moreover, meta-species inherit all reactions from their predecessors. It allows for an approach where one defines basic meta-species with reactions that act upon them and, in a second step, assembles them into a product meta-species with all combined reactions.

Unlike rule-based languages, MobsPy's object-oriented approach allows state queries that reference not only the species containing a given state but also all its inheritors simultaneously. Additionally, meta-species states can be queried dynamically in resulting data without the need to define observables. Moreover, MobsPy simplifies modeling reproduction-like reactions (e.g., $R \rightarrow 2R$, $2R \rightarrow 3R$ ) by automatically assigning states to unmatched meta-species based on the order of reactant-product transformations. In contrast, for this reaction type, rule-based languages require explicitly defining individual reactions for each state.

In contrast to BioCRNpyler's species, all reactions in MobsPy can be directly inferred from the model's code. Moreover, akin to the provided BioCRNpyler classes, meta-reactions in MobsPy can be encapsulated within Python functions, enabling sharing and reuse throughout different models.

MobsPy focuses on syntax simplification and readability, similarly to Antimony which focuses on a more readable alternative to SBML. However, Antimony does not provide reaction modularity and does not attempt to streamline the number of reactions in a model. Similarly to Antimony, MobsPy has mid-simulation model changes implemented.

MobsPy further incorporates the Pint Python module [25] to allow for units in rates, counts, and concentrations that are checked for consistency and appropriately converted for simulation. An example is Michaelis–Menten kinetics, where BioCRNpyler verifies units of both the Michaelis constant and the limiting rate. Additionally, it provides a means of specifying non-mass-action rates directly via arithmetic expressions in Python and automatically checks if the resulting unit is consistent.

For simulation, MobsPy internally generates SBML files and runs deterministic (ODE) or stochastic (Gillespie and Gibson–Bruck) simulations via the COPASI bindings from BasiCO. Simulation via other backends, such as libRoadrunner, is possible via SBML export. Additionally, the provided features available in the MobsPy Python module are: (i) Parametric sweeps and the utilization of multiple CPU cores when appropriate. Parameters, as well as meta-species, are automatically assigned names that are derived from their corresponding Python variables unless otherwise specified. It eliminates the burden of using strings to name them redundantly. Further, conducting a parametric sweep is effortlessly achieved by assigning multiple values to a parameter. (ii) Event handling, where MobsPy accommodates the addition of time-based or condition-based events that dynamically modify meta-species counts or parameter values during simulations. In particular, condition-based events offer the flexibility to end simulations under specific conditions. (iii) The possibility of enchaining simulations is expressed in MobsPy as a simple summation of simulation objects. Furthermore, each simulation within the concatenated one can be simulated either deterministically or stochastically. A particular use-case of this is simulation via the computationally more expensive stochastic solver until a specific condition is reached and then switch to a faster ODE solver.

## Design and implementation

### The MobsPy language

We discuss the main features of MobsPy along with a running toy example. Consider a system of two locations where trees grow (Fig 1a): one location where trees grow and reproduce within a dense population and one where they thrive in a sparser environment. Trees die and age, with young trees in the dense environment dying at a higher rate due to competition for resources and space. Further, a tree's leaf colors cyclically change from green to yellow to brown and back to green.

MobsPy simplifies the definition of BCRNs through a bottom-up approach. Base meta-species work as foundational building blocks that encapsulate elementary functions. The base meta-species are then combined to form more complex species, inheriting both reactions and states from their foundational predecessors. We start with modeling the aging process. A base meta-species named `Age` is introduced, and the reaction (») operator is applied to add the transition from a young `Age` to an old `Age` with a given reaction rate that is per-default mass-action and has a rate constant specified within subsequent brackets. The construct is now ready to be used for the specification of more complex meta-species via inheritance.

```
Age = BaseSpecies()
# The characteristics young and old are automatically added
     to Age
# by the dot (.) operator
Age.young >> Age.old [1/10/u.year]
```

Reversible reactions are expressed by applying the `Rev[]` operator to a reaction, followed by a forward and backward rate in brackets. New characteristics of a meta-species are introduced via the dot (`.`) operator. It can be done inside (like in the `Age` example) or outside of reactions. We will use the latter in our running example by adding characteristics to a `Colored` and a `Location` meta-species.

```
Colored, Location = BaseSpecies()
Colored.green, Colored.yellow, Colored.brown
Location.dense, Location.sparse
```

It is further possible to create meta-species and assign reactions to them without introducing new characteristics. As an example, consider a `Mortal` meta-species to model a death reaction. In the reaction, we make use of `Zero`, a special meta-species that represents the absence of reactants or products in a reaction.

Furthermore, MobsPy allows one to specify reaction rates that depend on the characteristics of the reactants by replacing the reaction rate constant with a function whose arguments are the reactants. In case the returned expression contains a reactant (e.g., in `lambda r1, r2: r1/(r1 + r2)`) it is taken as a reaction rate. Otherwise, as in the example below, mass-action kinetics is assumed and the returned value is taken as a reaction rate constant.

```
Mortal = BaseSpecies()
# Rates can be functions with reactants as arguments. Here,
    the rate is mass-action.
# It is non-zero only if the reactant is 'old'.
Mortal >> Zero [lambda r1: 1/u.year if r1.old else 0]
```

We are now ready to construct the `Tree` meta-species from the basic meta-species by combining the `Colored`, `Age`, `Mortal`, and `Location` meta-species through the multiplication (`*`) operator. The combination encapsulates the representation of leaf color, lifespan, mortality potential, and location (dense or sparse environment).

```
Tree = Age * Colored * Mortal * Location
```

The species within `Tree` are obtained via the Cartesian product over the sets of states of its predecessor meta-species. All characteristics of a predecessor species are accessible by a meta-species that inherits from this predecessor species. Illustrative is the view of a meta-species as a point in a multi-dimensional space (Fig 1b). A species in `Tree` has `Age`, `Colored`, and `Location` components with either 'young' and 'old' in the `Age` component, 'brown,' 'green,' or 'yellow' in the `Colored` component, and 'dense' or 'sparse' in the `Location` component. In our example, 12 species of the form `Tree.(young|old).(brown|green|yellow).(sparse|dense)` are subsumed within the `Tree` meta-species.

The dot operator has different semantics in reactants and products. When used in a meta-species that is a reactant, it acts as a query and refines the meta-species to those species that possess this characteristic. For example, the reactant `Tree.young` is the set of species of the form `Tree.young.(brown|green|yellow).(sparse|dense)`. Conversely, when used in products, the dot is used to specify a change of characteristics. For example, the product `Tree.young` is the respective reactant with characteristics age set to 'young'. Following these rules, in the running example, the meta-reaction `Age.young » Age.old` is compiled into the following reactions:

```
Tree.young.brown.dense-> Tree.old.brown.dense
Tree.young.brown.sparse-> Tree.old.brown.sparse
Tree.young.green.dense-> Tree.old.green.dense
Tree.young.green.sparse-> Tree.old.green.sparse
Tree.young.yellow.dense-> Tree.old.yellow.dense
Tree.young.yellow.sparse-> Tree.old.yellow.sparse
```

Since the dot operator on products only defines characteristics of the reactant that are to be changed, it is essential to specify which product originates (and thus inherits default characteristics) from which reactant. While a matching of products with reactants is unambiguous for reaction `S1.a » S1.b` with meta-species `S1`, matching in reactions with multiple products and reactants is *a priori* ambiguous. Starting with the first product, MobsPy cycles through the reactants until it finds one that has the same meta-species as the product upon which the product is matched with the reactant. It is continued with every product until all are matched with a reactant (Fig 2).

We make use of the cyclic matching in the subsequent two reactions of the running example.

```
# competition
Tree.dense.old + Tree.dense.young >> Tree.dense.old
     [1e-10*u.decimeter/u.year]
# replication
Tree.old >> Tree + Tree.young [0.1/u.year]
```

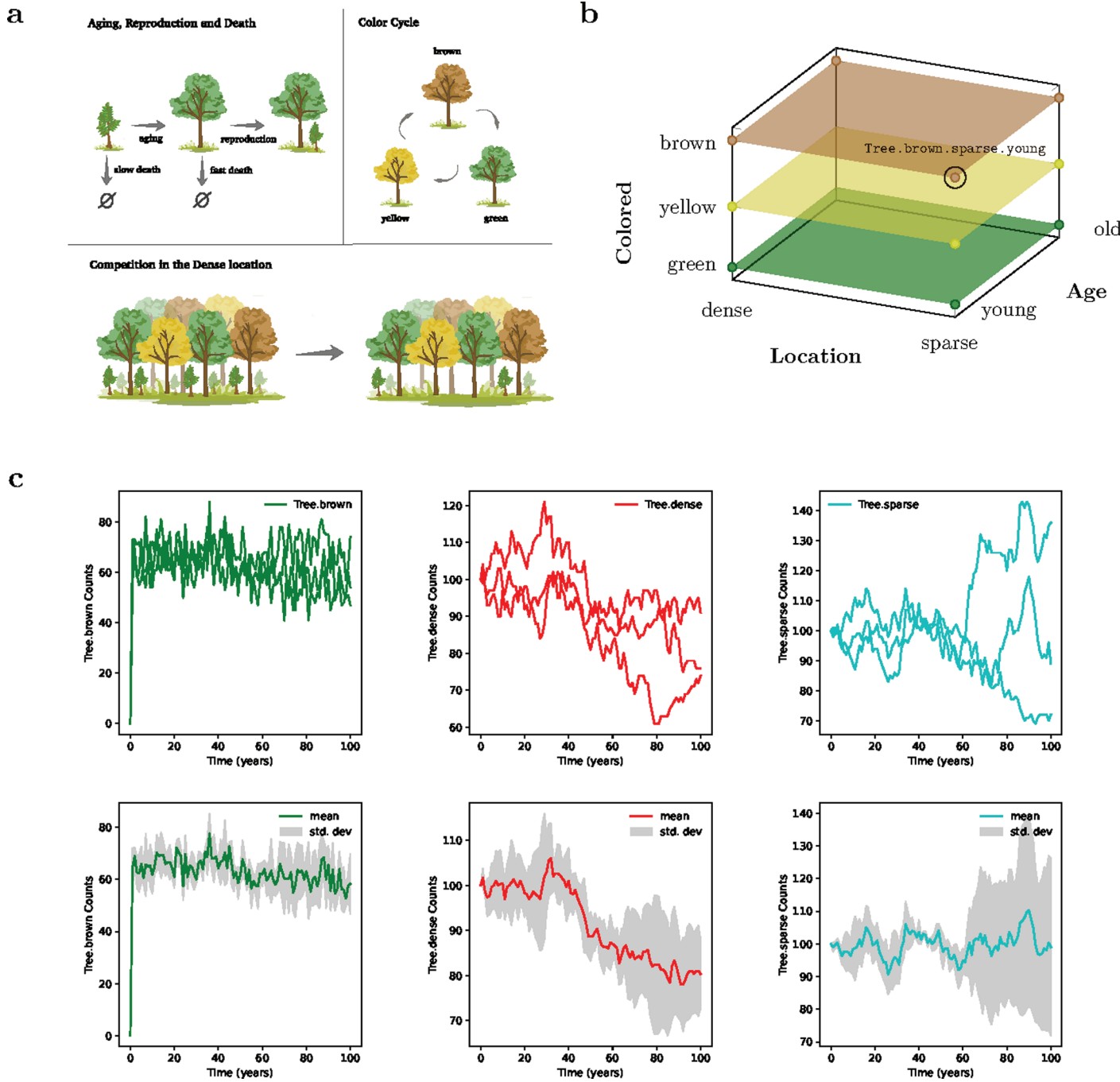

**Fig 1. (a) System dynamics: schematic representation of all the reactions in the Tree model, which are aging, reproduction, color cycle, and competition. (b)** Meta-species `Tree` created by multiplication of the three base-species `Age`, `Location`, and `Colored`. One of the twelve species generated, `Tree.brown.sparse.young`, has been labeled. **(c)** MobsPy default plots after simulating the Tree model (*n* = 3 stochastic runs). In the top row of the panel, individual runs are shown. The bottom plots depict the mean and standard deviation.

The first reaction models the behavior of old trees suppressing the growth of young trees within the dense location. The matching implies that the old `Tree` survives, as opposed to the young tree surviving and becoming old. The second reaction models the replication of old

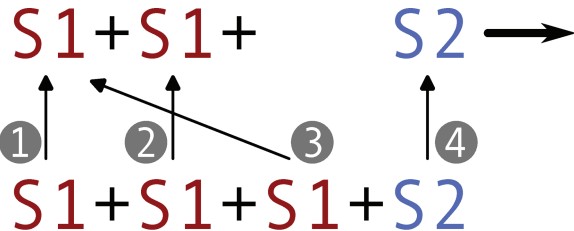

**Fig 2. Matching products with reactants for the example meta-reaction 2 * S1 + S2 » 3 * S1 + S2 where S1 and S2 are distinct meta-species.** MobsPy starts with the leftmost product and matches the leftmost reactant with the same meta-species (indicated by an arrow) in step 1. The process is repeated with the next reactant (step 2). For the next product (step 3), MobsPy cycles back to the first reactant. Finally both S1 are matched in step 4.

trees. Here, the old tree remains unchanged and creates a new tree that is identical to the old one, except being young.

The next reaction is a low-rate sexual reproduction reaction where two trees generate a new young one containing the characteristics of a parent. In this reaction, trees in the same location and color have a much higher rate for reproduction. This is archived by extracting the location of a reactant with the call operator using the name of a predecessor meta-species.

```
# A sexual reproduction reaction
bf = 1e-10 * u.decimeter ** 2 / u.year
rep_r = lambda t1, t2: (
    5 * bf
    if (Location(t1) == Location(t2) and Colored(t1)
        == Colored(t2))
    else bf
)
2 * Tree >> 2 * Tree + Tree.young[rep_r]
```

This meta-reaction adds 144 reactions to the model and shows how MobsPy's automatic round-robin order allocation can simplify the definition of reproduction-based reactions.

It remains to specify the cyclic color transition of the `Tree` species over time. This is achieved via the characteristics (`c`) function of a meta-species: `S.c(s)` is equal to `S.v`, where `v` is the value of the string variable `s`. We use this method to loop over the colors.

```
colors = ['green', 'yellow', 'brown']
for color, next_color in zip(colors, colors[1:] + colors[:1]):
Tree.c(color) >> Tree.c(next_color) [10/u.year]
```

For initial conditions, the call (`()`) operator is used to set initial counts of meta-species. In case the meta-species contains multiple species with different characteristics, the assignment is made to the default characteristics of the meta-species, which is the one that was added first. For example, the default characteristics of `Colored` is green, and the default characteristics of `Age` is young. Example initializations for the tree model are:

```
Tree.dense(50), Tree.dense.old(50), Tree.sparse(50),
    Tree.sparse.old(50)
```

MobsPy distinguishes between meta-species solely used to construct other meta-species and those that will be used in a simulation. For example, `Age`, `Colored`, and `Mortal` are only used to construct `Tree`, while the `Tree` meta-species contains the species whose behavior is to be simulated. In MobsPy, species that are to be simulated are passed to the `Simulation` constructor separated by `|`. The constructor then compiles these meta-species and the meta-reactions that act upon them into sets of species and reactions. For the running example, this is achieved via:

```
MySim = Simulation(Tree)
```

Finally, stochastic (Gillespie) or deterministic (ODE) simulations are run for a given duration, with units of the model being checked for consistency and default plots being displayed (Fig 1c).

```
# run 3 stochastic simulations over a 20-year duration
Tree.dense(50), Tree.dense.old(50), Tree.sparse(50),
    Tree.sparse.old(50)
MySim = Simulation(Tree)
MySim.run(volume=1 * u.meter ** 2, unit_x='year',
    duration=100 * u.years, repetitions=3, output_concentration
        =False,
    simulation_method='stochastic', save_data=False, plot_data
        =False)
MySim.plot_stochastic(Tree.dense, Tree.sparse, Tree.brown)
```

## Results

### Models

Next, we present MobsPy models from several BCRNs in the literature.

**Mutual annihilation protocol.** In some of the authors' previous work [26], we proposed a stochastic bacterial system for reconstructing and amplifying a chemical signal constituted of two species A and B. If A is in excess, the signal is logical 1, while an excess of B encodes a logical 0. Desired is a large gap between concentrations of A and B by a mutual annihilation reaction between A and B. Concurrently, the concentrations of both species are amplified by replication reactions.

We show a simulation of the system in two phases: one deterministic growth phase that establishes high concentrations of species A and B with the replication reactions and without the annihilation reaction, and a subsequent stochastic phase that enables the annihilation reaction. The first phase can be defined as follows:

```
from mobspy import
* import os

A, B = BaseSpecies()

# Replication reactions
A >> 2 * A [1.05 / u.h]
B >> 2 * B [1 / u.h]
```

```
# Initial counts
A(1 / u.ml), B(1 / u.ml)
# First phase
S1 = Simulation(A | B)
S1.duration = 3*u.h
S1.volume = 1*u.ml
```

The second phase is modeled by adding the annihilation reaction to meta-species `A` and `B` and creating a new simulation object `S2` with the additional reaction. The complete two-phase model `S` is obtained as the sum of `S1` and `S2`.

```
# Annihilation reaction
A + B >> Zero [0.1/u.h]

S2 = Simulation(A | B)
S2.duration = (A <= 0) | (B <= 0)
S2.method = 'stochastic'
S2.volume = 1*u.ml

# Sum both simulations
S = S1 + S2 S.unit_x,
S.unit_y = u.h, 1/u.ml
S.add_plot_params(A={'ylabel': "A (mL$^{-1}$)"}, B={'ylabel':
        "B (mL$^{-1}$)"},
   vertical_lines=[3], tight_layout=True,
   xlabel_fontsize=14, ylabel_fontsize=14,
   suptitle='Mutual Annihilation Protocol',
        suptitle_fontsize=18)
S.plot_config.save_to = os.path.dirname(os.path.dirname
        (os.path.abspath(__file__))) + \
   '/images/Mutual_Annihilation/Mutual_Annihilation.pdf'
S.repetitions = 10
S.run()
```

The results (Fig 3) are consistent with what is expected from our previous analysis: Since the second phase is stochastic, there is a non-zero, although small, probability that the concentration of `B` surpasses that of `A` and `B` survives even though it is in a lower concentration than `A` at the beginning of the second phase.

**Bistable gut inflammation detector.**   Riglar *et al.* [27] implemented a system designed to detect intestinal inflammation within the mammalian gut. It is composed of a trigger element and a memory element. The trigger element detects the presence of one of the gut inflammation byproducts (tetrathionate, `TTR`). The introduction of `TTR` results in the phosphorylation of `TtrS` which then triggers a chain reaction that up-regulates the production of `Cro`. The memory element is based on the interaction of two proteins, `Cro` and `CI`, which function as mutual repressors of each other, establishing a state of bistability where either `Cro` is high and `CI` low, or vice versa. The chemical species `Cro` then represses `CI`, going from a low-`Cro` high-`CI` state to a high-`Cro` low-`CI` state. After the state transition, the high-`Cro` low-`CI` state remains stable even if the inflammation byproduct `TTR` leaves the system. It returns to a low-`Cro` high-`CI` state if `CI` is added to the system. In MobsPy we define:

# Mutual Annihilation Protocol

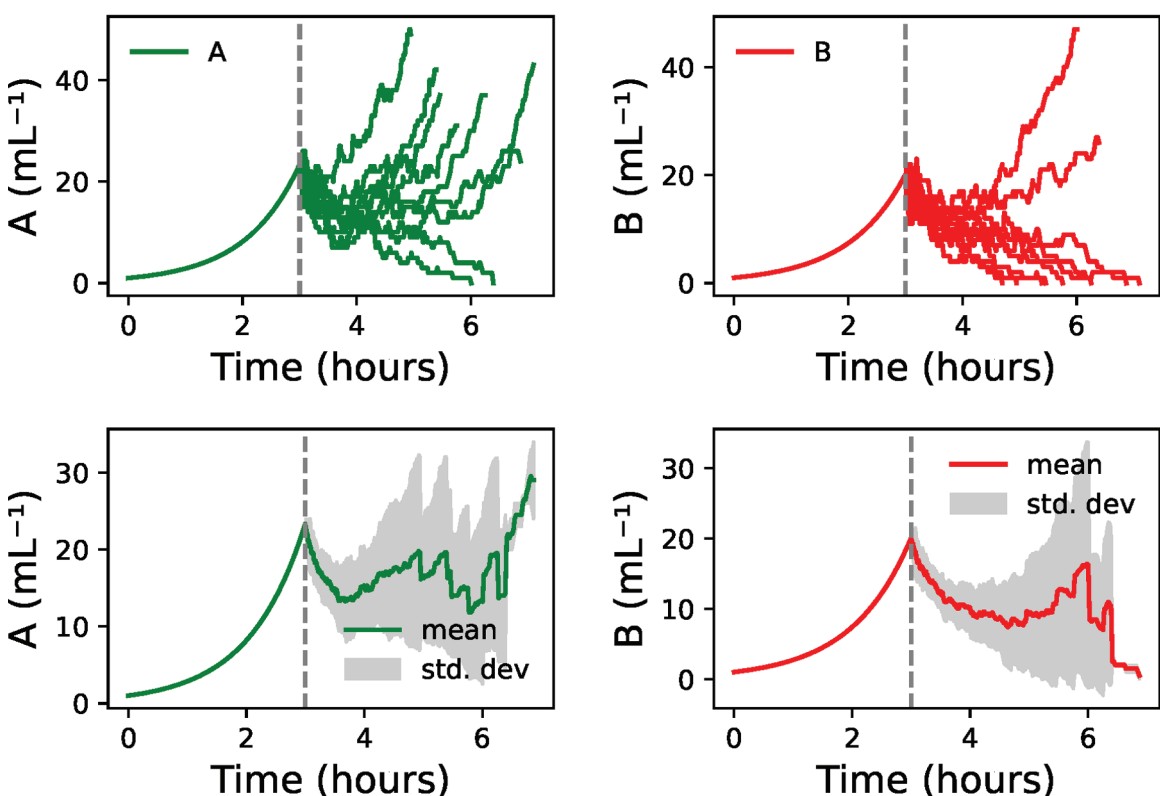

**Fig 3. Simulated dynamics of the Mutual Annihilation protocol [26].** By default, for stochastic simulations, MobsPy plots the runs of each species as well as the mean and standard deviation. The dotted gray line represents the transition from the deterministic only-growth phase to the stochastic annihilation. The species A survives in the majority of the runs.

```
from mobspy import *
import os

Dummy, Dilutable, Promoter, TTR, Phosphorable = BaseSpecies()
TF, Reporter = New(Dilutable)
TtrR, TtrS = New(Phosphorable)
Cro, CI = New(TF)
Pcro, PcI = New(Promoter)
Promoter.bound, Promoter.unbound, Phosphorable.dephospho,
    Phosphorable.phospho
```

The phosphorylation reactions that trigger the production of `cro` are specified via:

```
# TtrS triggers the state transition, TTR indicates the presence
    of gut inflammation
TtrS.dephospho + TTR >> TtrS.phospho + TTR [1/u.min]
TtrS.phospho + TtrR.dephospho >> TtrR.phospho + TtrS.dephospho
    [1/u.min]
```

```
TtrR.phospho >> TtrR.dephospho [5*1e-3*1/u.second]
TtrR.phospho >> Cro + TtrR.phospho [50*0.02*1/u.min]
```

To simulate the delayed addition of `CI`, we use a placeholder meta-species `Dummy`. Starting with an initial count of zero, the `Dummy` meta-species is set to one during the execution of the simulation, triggering an influx of `CI`.

```
# Dummy represents the introduction of a constant flow of CI in
        the system Dummy >> CI + Dummy [30*0.02*1/u.min]

# Dilution and degradation
Dilutable >> Zero [0.02/u.min]
TF >> Zero [lambda r1: 0 if r1.is_a(CI) else 1.6e-2*1/u.min]

def Expression(P, Pdt, rate_expression, rate_leaky): P >>
    P + Pdt [lambda r1: rate_expression if r1.unbound else
        rate_leaky]

def Repression (Prom, Rep, rate_binding, rate_unbinding):
    Prom.unbound + 2*Rep >> Prom.bound [rate_binding]
    Prom.bound >> Prom.unbound + 2*Rep [rate_unbinding]

# Pcro produces Cro, and PcI produces CI Expression(Pcro,
        Cro, 5/u.min, 0.05/u.min), Expression(PcI, CI,
            4.25/u.min, 0.05/u.min)
# Cro represses Pcro, and cI represses PcI Repression(Pcro,
        CI, 1, 50**2), Repression(PcI, Cro, 1, 40**2)
```

One can validate that the introduction of the inflammation byproduct results in the low-`Cro` state transitioning to a high-`Cro` state that remains after the event that removes the byproduct. MobsPy's event syntax uses the keyword `with` and the Simulation object with the methods `event_time` for time-based triggers or `event_condition` for conditional-based triggers. The species concentrations to be set in an event are stated inside the `with` block, similar to setting initial values.

```
model = set_counts({Pcro: 1, PcI: 1, Cro: 0, CI: 10,
    Reporter: 0, TtrR: 1, TtrS: 1, TTR: 0, Dummy: 0})
S = Simulation(model)
S.duration = 160*u.hour
# Event implementation
    with S.event_time(25*u.hour):
    TTR(1)
with S.event_time(60*u.hour):
    TTR(0)
with S.event_time(90*u.hour):
    Dummy(1)
with S.event_time(130*u.hour):
    Dummy(0)
```

```
S.plot_data = False S.unit_x,
S.unit_y = u.hour, 1/u.ml
S.step_size, S.a_tol = 0.01*u.hour, 1e-16
S.plot_config.title, S.plot_config.title_fontsize =
    'Toggle Switch', 16
S.plot_config.figsize = (6.5, 4)
S.plot_config.vertical_lines = [25, 60, 90, 130]
S.plot_config.ylabel = r"Conc. (mL$^{-1}$)"
S.plot_config.save_to = os.path.dirname(os.path.dirname(os.
        path.abspath(__file__))) + \
    '/images/Toggle_Switch/Toggle_Switch.pdf'
S.run()
S.plot(CI, Cro)
```

The simulation (Fig 4) shows that the introduction of the inflammation byproduct (at 25 hours) results in a transition from a low-`Cro` high-`CI` state to a high-`Cro` low-`CI` state, with the latter state persisting after the inflammation marker removal (at 60 hours). Moreover, the reintroduction of `CI` results in a transition back to the low-`Cro` high-`CI` state (at 80 hours).

**Phage transmission system.**   Pathania *et al.* [5] studied a bacterial system that utilizes bacteriophages for transmitting antibiotic resistance. In their work, a bacterial population (`Donor`) secretes phages carrying an antibiotic resistance gene, which infects a bacterial population (`Receiver`) devoid of such.

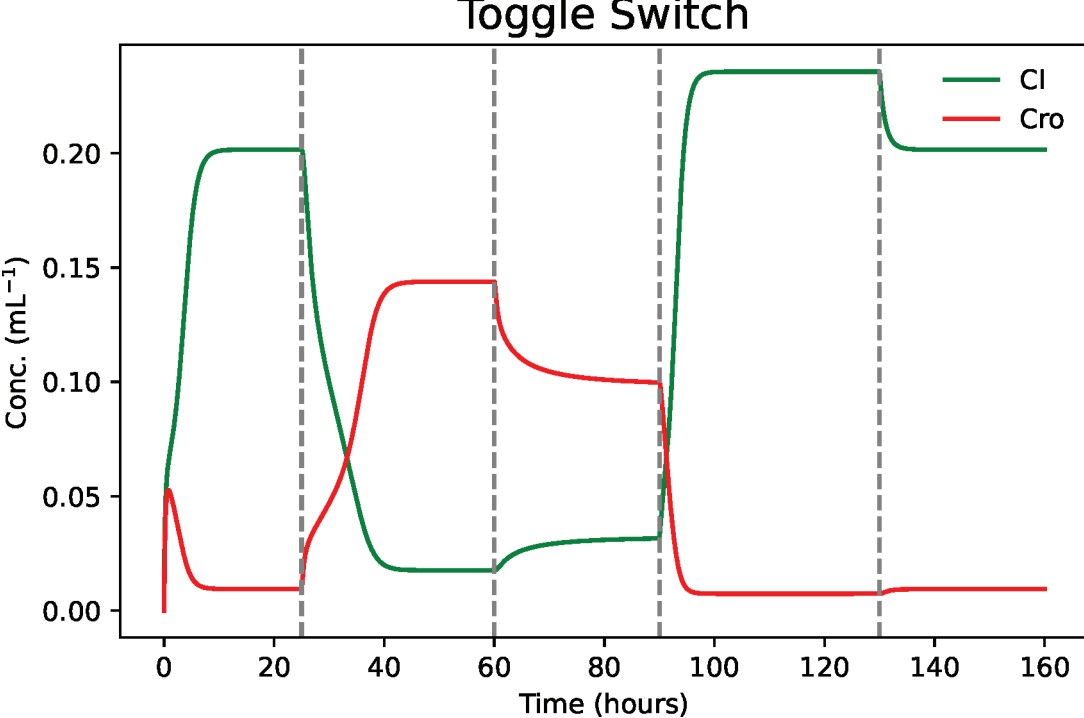

**Fig 4. Results obtained from the Bistable Gut Inflammation Detector model.** The exposure of the system to `TTR` (at 25 hours) shifts the system to a high-`Cro` state, which remains even after the removal of `TTR` (at 60 hours). The system only returns to a low-`Cro` state when introducing `CI` (at 80 hours).

Cell duplication was modeled using two distinct resources (R1 and R2). Resources R1 and R2 are consumed for bacterial replication with a faster uptake rate for R1. Further, R1 is also the only resource consumed by a Donor for phage production. The discrepancy in consumption rates leads to the depletion of R1 with only R2 being available afterward.

Like the Mutual Annihilation protocol, this experiment is sectioned into two phases with antibiotics being introduced in the second phase. If Donor species can produce sufficient phages to transmit the antibiotic resistance to the Receiver species, the Receiver species will survive the introduction of antibiotics; otherwise, they die.

```python
from mobspy import *
volume = 1*u.microliter
c_donor, c_rec = 10000*volume/u.microliter, 1000*volume/
    u.microliter
c_r1, c_r2 = 3*9e6*volume/u.microliter, 3e8*volume/u.microliter
Mortal, Resource, Age, Dead = BaseSpecies()
Phage = New(Mortal)
Donor = New(Mortal * Age)
Receiver = New(Age)
R1, R2 = New(Resource)

Age.young >> Age.old [1/4*(1/u.min)]

# Receiver
rm = lambda r: 1/c_r1 if r.is_a(R1) else 1/c_r2
cm = lambda r: 1/c_donor if r.is_a(Donor) else 1/c_rec
grw_r = lambda r1, r2: 1/20*cm(r1)*rm(r2)*(u.l/u.s) \
    if r2.is_a(R1) else 0.08/20*cm(r1)*rm(r2)*(u.l/u.s)
inf_r = lambda r1, r2: 10*3e-11*cm(r1)*rm(r2)*(u.l/u.s) \
    if r1.old else 0.004*10*3e-11*cm(r1)*rm(r2)*(u.l / u.s)
Receiver.not_infected + Phage >> Receiver.early_infection [inf_r]
Receiver.early_infection >> Receiver.late_infection
    [1 / (3 * u.min)]
```

In the model, a Phage can bind a dead Receiver, resulting in the loss of a phage. We keep track of dead Receiver species through a reaction transforming a Receiver into Dead.

```python
# Receiver replication and death
Receiver.old + Resource >> Receiver.young + Receiver.not_infected.
    young [grw_r]
Receiver >> Dead [1e-4 / u.s]

# Donor Reactions
phage_rate = lambda r1, r2: cm(r1)*rm(r2)*850*(u.l/u.s) if r2.
    is_a(R1) else 0
Donor.old + Resource >> 2*Donor.young [lambda r1, r2: grw_r(r1,
    r2)/2]
Donor + Resource >> Donor + Phage + Resource[phage_rate]
```

```
# Death Reactions
Mortal >> Zero [lambda r: 1e-4*cm(r)*1/u.s if r.is_a(Donor)
    else 0.074*cm(r)/24*(1 / u.min)]
Dead + Phage >> Dead [lambda r1, r2: 0.074/24*cm(r1)*rm(r2)*
    (u.l/u.min)]

model = set_counts({Receiver.not_infected: c_rec, Donor: c_donor,
    Phage: 0, R1: c_r1, R2: c_r2, Dead: 0})
S1 = Simulation(model)
S1.duration = 2000*u.s

S1.volume = volume
S1.plot_data = False
```

In the second phase `Antibiotics` is introduced into the system:

```
Antibiotics = BaseSpecies()
Antibiotics + Receiver.not_infected >> Zero
    [0.015*1/1e9*(u.l/u.s)]

# Change to concentration - Is currently in counts
    Antibiotics(1e7 / u.microliter)
S2 = Simulation(model | Antibiotics)
S2.duration = 3*u.hours
S2.volume = volume
Sim = S1 + S2
Sim.unit_x = u.h
Sim.unit_y = 1/u.ml
Sim.run()
Sim.plot_raw('plot_config_donor.json')
```

The simulation results (Fig 5) are in accordance with the original work: One observes that the `Receiver` population has survived the introduction of `Antibiotics` due to a successful transmission of antibiotic resistance.

**Synchronized cycles of bacterial lysis.** Omar Din *et al.* [28] designed a bacterial system for *in vivo* drug delivery through the synchronized death of a bacterial population. Each bacterium produces *N*-acyl homoserine lactone (`AHL`) and secretes it into the medium. Consequently, the concentration of `AHL` increases as the number of bacteria increases. If the concentration of `AHL` passes a threshold, it triggers a lysis process via quorum sensing that kills most of the bacterial population, leaving only a slim fraction compared to its pre-lysis size. This process creates a cyclic behavior.

Model-wise, meta-species `LuxI` is a protein that dictates the secretion rate of `AHL` by a cell. Meta-species `Lysis` models the intracellular enzyme responsible for the lysis process. This enzyme inhibits peptidoglycan biosynthesis [29], causing cellular death. The production of `Lysis` increases as a consequence of an increase in the `AHL` concentration. Bacterial cells are modeled by the meta-species `Cell`.

```
from mobspy import *

n_0, lysis_0, k, b, mu_g, mu, c_l, alpha_0, alpha_h, AHL_0,
```

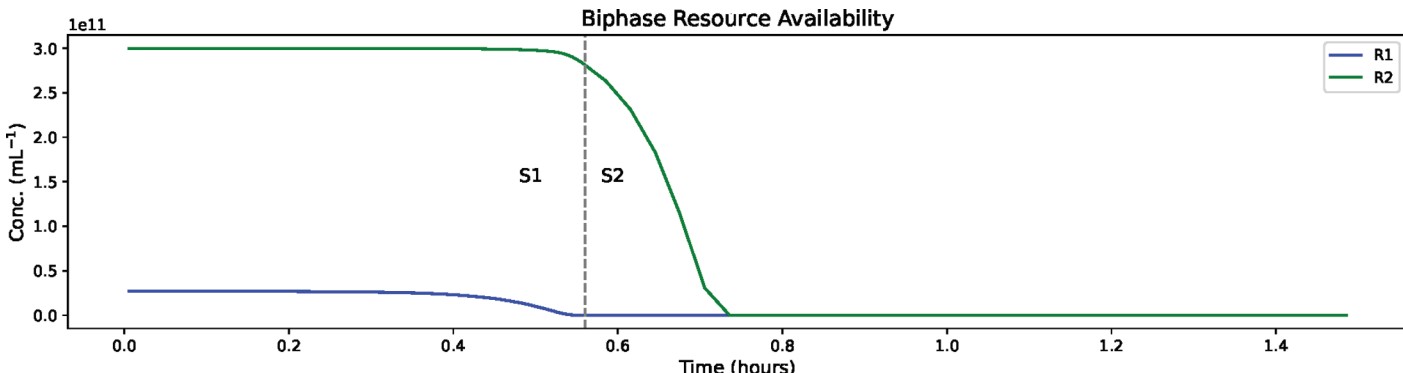

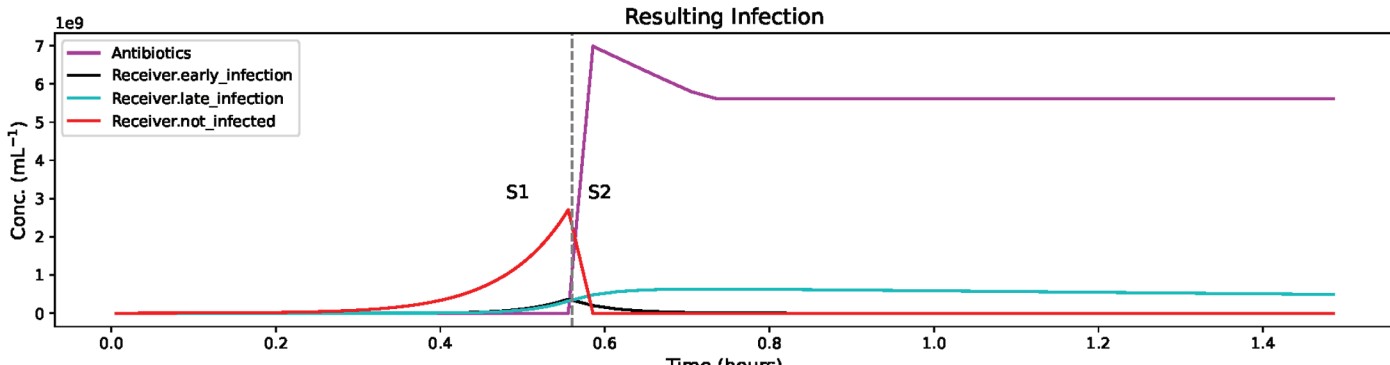

**Fig 5. Simulation results for the phage communication circuit from Pathania *et al.* [5].** The gray line separates between the simulation phase dedicated to resource dynamics (`S1`) and the simulation phase with an antibiotic introduction (`S2`) **Top** The resource availability achieved in this model. Resource `R1` is consumed faster than `R2`, resulting in a window of time where mostly `R2` is available, and `R1` is depleted. **Bottom** Resulting infection progress. Only infected `Receiver` species remain.

```
gamma_l, c_i, \ gamma_i, gamma_c = (10, 2, 10/u.h, 25/u.h,
0.2/u.h, 12/u.h, 0.5, 0.5/u.h, 35/u.h, 5, 2/u.h,
1, 2/u.h, 12/u.h)

# N, L, H, I - Are the respective counterparts of the
    meta-species below
# in the original paper's model
Cell, Lysis, AHL, LuxI = BaseSpecies()
```

The BCRN does not exclusively use mass-action-kinetic reaction rates. In MobsPy this can be expressed by Python functions that return expressions that include reactants (see the following example). Such expressions are compatible with units and checked for consistency.

```
# Cell related reactions
Cell >> 2*Cell [lambda cell: mu_g*cell*(n_0 - cell)]
# Inner-cellular Lysis enzyme attacks the cell membrane,
    leading to cell death
Lysis + Cell >> Zero [lambda lysis, cell: k*cell/(1 +
```

```
                       (lysis_0/lysis)**2)]

# AHL related reactions
Cell + LuxI >> AHL + Cell + LuxI [b]
AHL + Cell >> Cell [lambda ahl, cell: mu*ahl/(1 + cell/n_0)]

# Lysis related reactions
AHL >> AHL + Lysis [lambda ahl: c_l*(alpha_0 + alpha_h*(ahl/AHL_0)
    **4/(1 + (ahl/AHL_0)**4))] Lysis >> Zero [gamma_l + mu_g]

# LuxI related reactions
AHL >> AHL + LuxI [lambda ahl: c_i*(alpha_0 + alpha_h*(ahl/AHL_0)
    **4/(1 + (ahl/AHL_0)**4))] LuxI >> Zero [gamma_i +
             mu_g + gamma_c]

Cell(5/u.l), Lysis(0/u.l), AHL(1e-5/u.l), LuxI(1e-5/u.l)

MySim = Simulation(Cell | Lysis | AHL | LuxI)
MySim.save_data = False
MySim.duration = 10*u.hour
MySim.unit_x, MySim.unit_y = u.hour, 1/u.ml
MySim.plot_config.title = 'Synchronized Bacterial Lysis'
MySim.plot_config.title_fontsize = 18 MySim.plot_config.
    xlabel_fontsize, MySim.plot_config.ylabel_fontsize = 14, 14
MySim.plot_config.ylabel = r"Conc. (mL$^{-1}$)"
MySim.plot_config.save_to = os.path.dirname(os.
    path.dirname(os.path.abspath(__file__))) + \
    '/images/Lysis_Clock/Lysis_Clock.pdf'
MySim.run()
```

Omar Din *et al.* [28] experimentally showed that the number of cells oscillate over time, which is reproduced by the above model (Fig 6).

**Logical XOR gate.** Tamsir *et al.* [11] implemented an XOR gate across three bacterial colonies, each colony functioning as a NOR gate (Fig 7). In the BCRN model, the XOR and NOR gates take the concentration of two species as inputs and produce another species as output. NOR gate C1 takes inputs `Ara` and `aTc`, outputting `AHL`. NOR gates C2 and C3 both take `AHL` as input, with C2's second input being `Ara` and C3's second input being `aTc`. The outputs of C2 and C3, named `OC12`, serve as input for a buffer that outputs yellow fluorescent protein `YFP`.

We define a meta-species `Location`, with characteristics `c1`, `c2`, `c3`, and `c4` representing the location of the three colonies and the buffer. The NOR gates are fixed in place; in contrast, the BCRN species `Ara`, `aTc`, `AHL`, and `OC12` can move between colonies, serving as inputs to subsequent gates. We model this with a species `Movable` that inherits from `Location` and that contains location transitions.

```
from mobspy import *
import numpy as np

max_plas, max_pbad, max_ptet = (1, 1, 1)
```

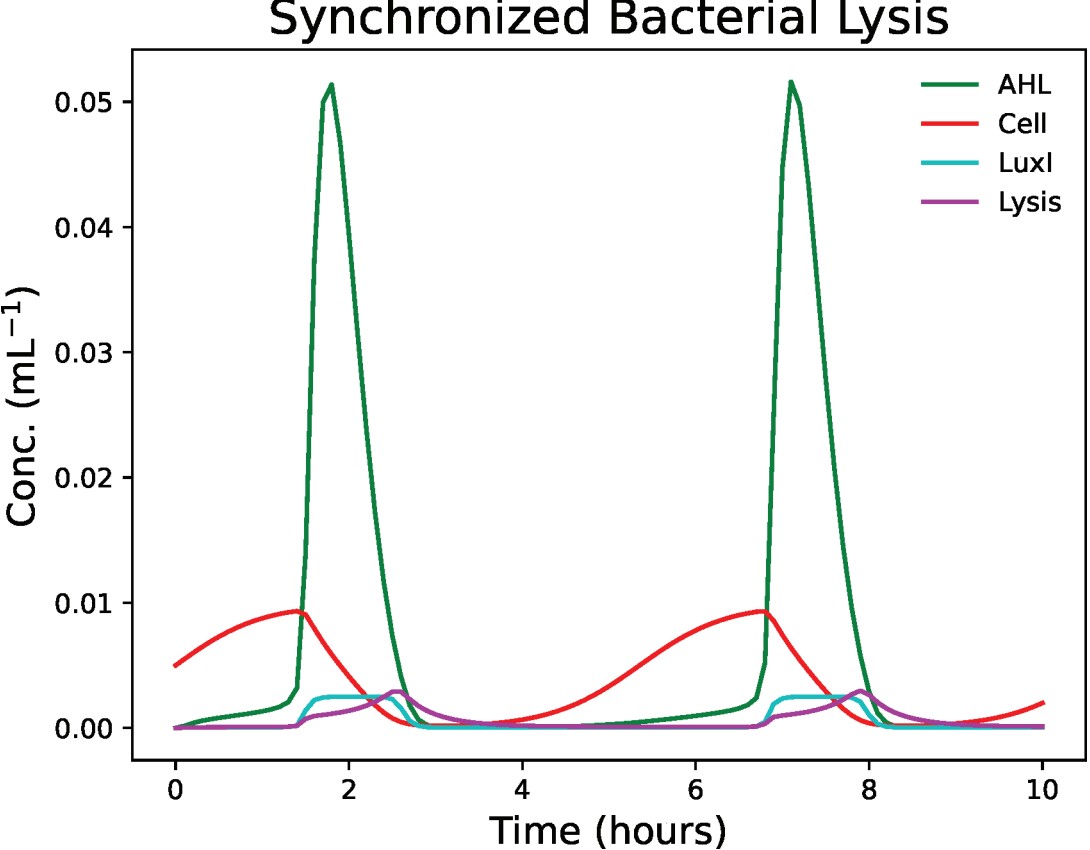

**Fig 6. Simulation results of the quorum sensing circuit by Omar Din *et al*. [28].** The simulation shows a synchronized lysis of bacteria. As the AHL levels and population size increase, eventually, a threshold is reached, and the lysis process starts killing the majority of the population. The cycle then repeats.

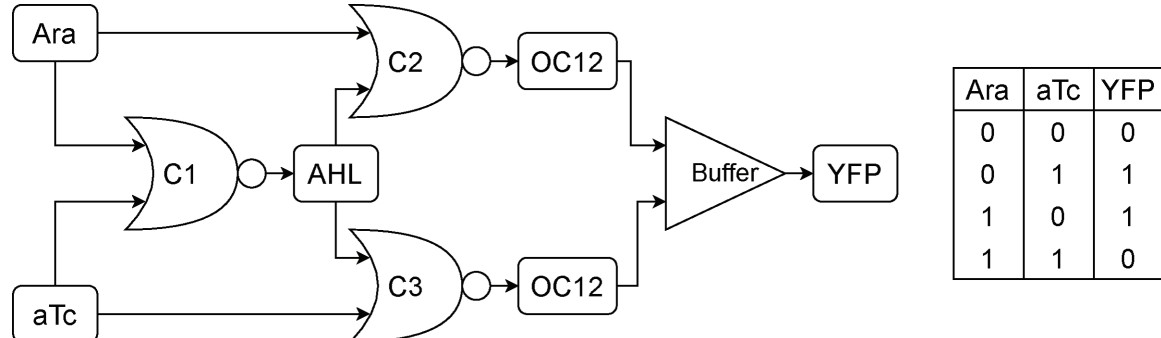

**Fig 7. Left** schematic of the XOR gate implementation [11]. **Right** Truth table of the circuit, equivalent to the specification of a XOR gate.

```
Mortal, Location = BaseSpecies()
Location.c1, Location.c2, Location.c3, Location.c4
PBad, PTet, Pcl, PLas, Movable = New(Location)
```

```
Ara, aTc, Cl, YFP, AHL, OC12 = New(Mortal*Movable)

# Death and movement reactions
Mortal >> Zero[1]
for x, y in zip(['c1', 'c1', 'c2', 'c3'], ['c2', 'c3',
    'c4', 'c4']):
  Movable.c(x) >> Movable.c(y) [0.1]
```

In this model, we measure the `YFP` output concentration for different input concentrations through a parametric sweep using the `ModelParameters` constructor. If a parameter has multiple values, MobsPy will run a parametric sweep using all values. Moreover, if multiple parameters have multiple values, MobsPy sweeps all possible combinations. In the model below, `ara_r` and `atc_r` are examples, controlling the concentration of `Ara` and `aTc` respectively.

```
n = 50
# Parametric sweeps for XOR gate analysis
ara_r, atc_r = ModelParameters([float(x) for x in np.linspace(0,
    35, n)],
        [float(x) for x in np.linspace(0, 2000, n)])

# Represents the entrance of aTc and Ara in the system
def diffusion_in_cell(Molecule, rate, locations):
  for l in locations:
    Zero >> Molecule.c(l) [rate]
diffusion_in_cell(Ara, ara_r, ['c1', 'c2'])
diffusion_in_cell(aTc, atc_r, ['c1', 'c3'])
```

The function `promoter_activation` represents the production of a BCRN species. It takes arguments such as the `Promoter` (the chemical species binding to NOR gate inputs for production), the `Ligand` (a NOR gate input), constants regulating input binding `tf_max`, `n`, and `K_d`, an expression controlling production rate `protein_production_rate`, and a list of locations where this reaction occurs `locations`. The third argument (`Protein`) is the species produced.

```
# Promoter activation function
def promoter_activation(P, Ligand, Protein, tf_max, n,
        K_d, protein_production_rate, locations):
tf_linked = lambda lig: tf_max*lig**n/(lig**n + K_d**n)
tf_free = lambda lig: tf_max - tf_max*lig**n/(lig**n + K_d**n)
pr = lambda r1, r2: protein_production_rate(r1, tf_linked(r2),
    tf_free(r2))
for l in locations:
  with Location.c(l):
    P + Ligand >> P + Ligand + Protein [pr]
```

The advantage of `promoter_activation` is that `Ara`, `aTc`, and `AHL` undergo similar reactions within their respective bacterial colonies. However, these reactions involve different rate expressions, constants, and locations. MobsPy retains similar aspects within the function definition, and the differences can be passed as function arguments.

The function `inverter_wire` represents the NOT part of the NOR gate. The first argument (`P`) is a meta-species that binds to the meta-species in the second argument (`R`) to repress the production of the meta-species in the third argument (`Signal`). When the concentration of R is high, `Signal` will not be produced, and when the concentration is low, `Signal` will be produced.

```
# Inverter for each nor gate
def inverter_wire(P, R, Signal, locations):
  rate_f = lambda r1, r2: 181*r1*350/(1 + 350 + 15*r2 + 50*r2
    + 15*50*0.18*r2**2)
  for l in locations:
     with Location.c(l):
        P + R >> P + R + Signal [rate_f]

# Custom buffer for clear visibility
def buffer(L, Signal, l, n, K):
   with Location.c(l):
      L >> L + Signal [lambda r: 30*r**n/(r**n + K**n)]
```

We finish the model by passing the arguments to the respective functions:

```
# Each promoter expression is written here and assigned to
    the promoter function pbad_p_rate = lambda r1, r2, r3:
    765*r1*(0.009 + 37.5*r2)/(1 + 0.009 + 37.5*r2 + 3.4*r3)
    promoter_activation (PBad, Ara, Cl, max_pbad, 2.8, 90,
    pbad_p_rate, ['c1', 'c2']) ptet_p_rate = lambda r1,
    r2, r3: 300*r1*350/(1 + 350 + 2*160*r3 + 160**2*r3**2)
    promoter_activation(PTet, aTc, Cl, max_ptet, 1.0,
    250, ptet_p_rate, ['c1', 'c3']) plas_p_rate = lambda
    r1, r2, r3: 69*r1*(0.002 + 100*r2)/(1 + 0.002 +
    100*r2) promoter_activation(PLas, AHL, Cl,
    max_plas, 1.4, 0.2, plas_p_rate, ['c2', 'c3'])
 # Inverter wires in c1, c2, and c3 inverter_wire(Pcl, Cl, AHL,
      ['c1']), inverter_wire(Pcl, Cl, OC12, ['c2', 'c3'])
 # Buffer for YFP buffer(OC12, YFP, 'c4', 4, 0.04)

# Count assignment and Sim
model = set_counts({Ara: 0, aTc: 0, Cl: 0, YFP: 0, PBad.c1: 1,
    PBad.c2: 1, PTet.c1: 1, PTet.c3: 1, Pcl.c1: 1, Pcl.c2: 1,
    Pcl.c3: 1, PLas.c2: 1, PLas.c3: 1, AHL: 0, OC12: 0})
S = Simulation(model)
S.duration = 100
S.plot_data = False
S.run()
```

The simulated data (Fig 8) clearly shows the characteristics of a XOR gate.

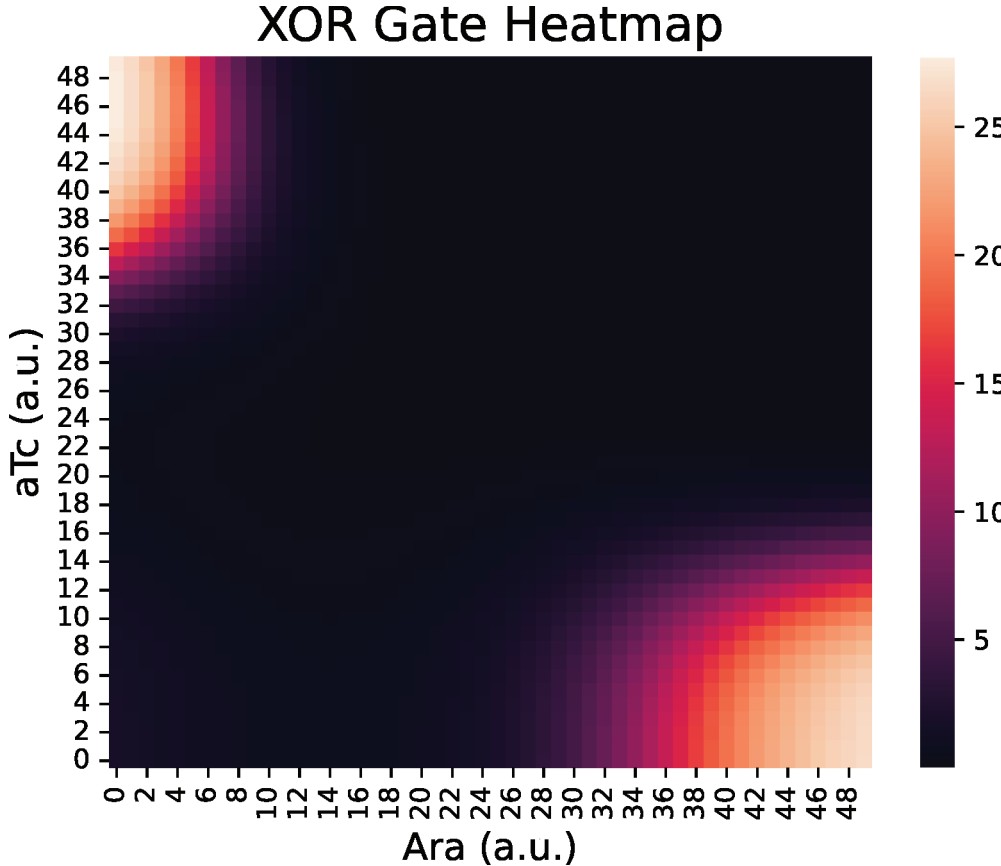

**Fig 8. Heatmap showing the simulated output of the XOR circuit by Tamsir *et al.* [11].** Only the regions with low-aTc and high-Ara as well as high-aTc and low-Ara show a high output as expected from a XOR gate.

## Comparison with related simulation frameworks

*Reaction modularity.*

MobsPy targets the concept of reaction modularity, i.e., a means to represent sets of reactions rather than individual reactions. The inspiration for this concept is drawn from the programming principle of modularity and object-oriented programming [30], where developers construct reusable code modules and classes with the goal of faster and less error-prone software development. By adopting a modular approach, both in programming and in the realm of BCRN modeling, the redundancy of re-implementing code or reactions is mitigated. This practice offers substantial benefits, including facilitating updates, where modifications are required only in one module, and enhanced debugging through a reduction in the number of lines of code or reactions that necessitate inspection.

Achieving reaction modularity within BCRN models is pursued through a number of languages such as Kappa [13,31], BioNetGen [14], and PySB [8]. In these languages, modularity is provided through a rule-based system that emphasizes the interactions between binding sites or domains within chemical species. Notably, chemical species sharing similar domains or binding sites are governed by the same rules, thus enabling the definition of multiple reactions through these shared rules. Thus, updating several reactions becomes as simple as changing a rule.

MobsPy focuses on an object-oriented approach where states and reactions are shared through inheritance. Although multi-state molecules are already used in STOCHSIM [22], BioNetGen [21], and BIOCHAM [32], they do not have the benefit of inheritance. The inheritance allows for multi-species queries of all species that share inheritors simultaneously.

BioCRNpyler [9] takes a different approach than modularity via binding sites. It uses mechanisms, components, and mixtures to achieve reaction modularity. Components can encompass several species, and each species can be of multiple components. Mechanisms define the reaction schema that applies to components, while mixtures establish the modeling context by constructing CRNs using components and mechanisms. BioCRNpyler's models rely on an extensive library of predefined Python classes. Notably, the components each mechanism interacts with are not explicitly presented in the model's code, requiring users to look into the assignment of reactions from the Python classes or to decipher proposed models from code. Moreover, extensions beyond the default classes are challenging, as the user must implement their modules using object-oriented programming. The method of rate definitions provided in an extension is also constrained to mass-action, Hill expressions, and similar options, with no provision for writing custom rate expressions [10].

For the simulation perspective of tools with reaction modularity, it is worth noting that out of the previously cited tools with reaction modularity, Kappa and BioNetGen are capable of simulating models created within their frameworks. On the other hand, BioCRNpyler and PySB, generate SBML files that can be integrated with dedicated simulation tools.

*Physical units.*

Like reaction modularity, the use of physical units helps reduce errors in complex models. It is particularly the case within the domain of synthetic biology, where unit systems play a central role [33] due to the interplay of many components from potentially different labs. Indeed, the iGEM registry, a registry used for genetic data in synthetic biology, presents all their rates with units when applicable [34]. Current tools that provide reaction modularity, like Kappa, BioNetGen, and BioCRNPyler [10,31,35], rely on the modeler to ensure unit consistency. MobsPy assists with unit consistency by automatically converting rates, concentrations, and counts to a standardized unit system. Additionally, when multiple parameters with units are used in a rate expression, MobsPy verifies unit consistency by performing automatic unit operations to ensure the resulting expression has coherent units.

In the rest of this section, we compare MobsPy with Kappa and BioCRNpyler in depth at the hand of examples from their tutorials rewritten in MobsPy.

**Comparison to Kappa.** We take the first model in the Kappa tutorial [31] (we removed the definition of observations for brevity). The model involves phosphorylation of the binding sites of a BCRN species C. The phosphorylation occurs twice, firstly when a species C reacts with A bound to a species B, and again when C reacts with A alone:

```
/* Signatures*/
%agent: A(x, c)             // Declaration of agent A
%agent: B(x)                // Declaration of agent B
%agent: C(x1{u p}, x2{u p})// Declaration of agent C with
                                2 modifiable sites
/* Variables */
%var: 'on_rate' 1.0E-4   // per molecule per second
%var: 'off_rate' 0.1     // per second
%var: 'mod_rate' 1       // per second
/* Rules */
```

```
//A and B bind and dissociate
'a.b' A(x[.]),B(x[.]) <-> A(x[1]),B(x[1]) @ 'on_rate', 'off_rate'
//AB binds unphosphorilated C
'ab.c' A(x[_], c[.]),C(x1{u}[.]) -> A(x[_], c[2]),C(x1{u}[2]) @
    'on_rate'
//ABC modifies x1 'mod x1' C(x1{u}[1]),A(c[1]) -> C(x1{p}[.]),
    A(c[.]) @ 'mod_rate'
//A binds x1_phos C on x2
'a.c' A(x[.],c[.]), C(x1{p}[.], x2{u}[.]) ->
    A(x[.],c[1]), C(x1{p}[.], x2{u}[1]) @ 'on_rate'
//AC modifies x2
'mod x2' A(x[.], c[1]),C(x1{p}[.], x2{u}[1]) ->
    A(x[.], c[.]),C(x1{p}[.], x2{p}[.]) @ mod_rate
/*Initial conditions */
%init: 1000 A(),B()
%init: 10000 C(x1{u}, x2{u})
```

As Kappa revolves around the occupancy of binding sites of a BCRN species, we represent the binding sites through meta-species `Link1` and `Link2`. The meta-species `Link1` is reserved for interactions involving A and B, while `Link2` is reserved for interactions involving A and C. Both `Link1` and `Link2` meta-species have two characteristics, not linked (`nl_2` and `nl_1`) and linked (`l_2` and `l_1`). Further, the meta-species `Phos` represents the phosphorylation states.

```
from mobspy import *

# We start with all base species
Link1, Link2, Phos_2 = BaseSpecies()
Link1.nl_1, Link1.l_1, Link2.nl_2, Link2.l_2
Phos_2.not_phos, Phos_2.phos_1, Phos_2.phos_2

# A and B can bound
A = Link1*Link2
B = New(Link1)

# C can be phosphorylated twice and bound to A
C = Phos_2*Link2

# A bounds to B, and they can unbind
Rev[A.nl_1 + B.nl_1 >> A.l_1 + B.l_1] [1e-4, lambda r: 0.1*r]

# A bound to B binds to C
A.nl_2.l_1 + C.nl_2.not_phos >> A.l_2.l_1 + C.l_2.not_phos
        [1e-4]

# C is phosphorylated and releases A
C.l_2.not_phos + A.l_2.l_1 >> C.nl_2.phos_1 + A.nl_2.l_1
    [lambda r: 1*r]
```

```
# C phosphorylated binds to unbound A
A.l_2.nl_1 + C.nl_2.phos_1 >> A.l_2.nl_1 + C.l_2.phos_1 [1e-4]

# C is phosphorylated again
A.l_2.nl_1 + C.l_2.phos_1 >> A.nl_2.nl_1 + C.nl_2.phos_2
    [lambda r: r]

# initial conditions
A(1000), B(1000), C(10000)
S = Simulation(A | B | C)
print(S.compile())
```

Furthermore, the Kappa graph-based algorithm also alleviates the burden of state space explosion by using graph connections instead of states for representation, while MobsPy simply focuses on syntax simplification. For instance, when representing the links of polymers in reactions, MobsPy would still be capable of writing the model. However, compilation would require a larger amount of time.

**Comparison to BioCRNpyler.** We next implement the two initial models from BioCRNpyler GitHub repository [10]. The first basic example is:

```
from biocrnpyler import *

# Species
A = Species("A")
B = Species("B")
C = Species("C")
D = Species("D")

#Reaction Rates
k1 = 3.
k2 = 1.4

#Reaction Objects
R1 = Reaction.from_massaction([A], [B, B], k_forward = k1)
R2 = Reaction.from_massaction([B], [C, D], k_forward = k2)

#Make a CRN
CRN = ChemicalReactionNetwork(species = [A, B, C, D],
    reactions = [R1, R2])
print(CRN)
```

The initial model consists of a simple BCRN where a BCRN species A reacts alone, resulting in two of species B, and B reacts alone, resulting in species C and D.

```
from mobspy import *

A, B, C, D = BaseSpecies()

A >> 2*B [3]
```

```
B >> C + D [1.4]

S = Simulation(A | B | C | D)
print(S.compile())
```

Unlike the model provided in BioCRNpyler's GitHub [10], MobsPy's automated variable naming eliminates the step of naming a BCRN using a string equal to its variable name. Additionally, in BioCRNpyler, to define reactions with mass action kinetics rates, the user must call the method `from_massaction` from the class `Reaction`. MobsPy's expressions and default mass-action kinetics make this step more succinct.

The second example showcases a more complex network:

```
from biocrnpyler import *

# Define a set of DNA parts
# this is a promoter repressed by tetR and has a leak reaction
ptet = RegulatedPromoter("ptet", ["tetr"], leak=True)
# constitutive promoter
pconst = Promoter("pconst")
# the Combinations A and B or just A or just B be transcribed
pcomb = CombinatorialPromoter(
    "pcomb",
    ["arac", "laci"],
    leak=False,
    tx_capable_list=[["arac"], ["laci"]]
    )
# regular RBS
utr1 = RBS("UTR1")
# regular RBS
utr2 = RBS("UTR1")
# a CDS has a name and a protein name. so this one
        is called GFP and
# the protein is also called GFP
gfp = CDS("GFP", "GFP")
# you can say that a protein has no stop codon.

fusrfp = CDS("fusRFP", "RFP", no_stop_codons=["forward"])

# regular RFP
rfp = CDS("RFP", "RFP")
# cfp
cfp = CDS("CFP", "CFP")
# a terminator stops transcription
t16 = Terminator("t16")

# Combine the parts together in a DNA_construct with
        their directions
construct = DNA_construct([
    [ptet, "forward"],
```

```
        [utr1, "forward"],
        [gfp, "forward"],
        [t16, "forward"],
        [t16, "reverse"],
        [rfp, "reverse"],
        [utr1, "reverse"],
        [pconst, "reverse"]
])

# some very basic parameters are defined
parameters = {"cooperativity": 2, "kb": 100, "ku": 10, \
              "ktx": .05, "ktl": .2, "kdeg": 2, "kint": .05}

# Place the construct in a context (TxTlExtract models a
    bacterial lysate
# with machinery like Ribosomes and Polymerases modeled explicitly)
myMixture = TxTlExtract(name="txtl", parameters=parameters,
    components=[construct])

# Compile the CRN
myCRN = myMixture.compile_crn()
print(myCRN.pretty_print(show_rates=True, show_keys=True))
```

For this model, BioCRNpyler uses an automated reaction rate assignment strategy that relies on the names of the species. Such a strategy is not used in MobsPy. Thus, we set all rates arbitrarily and focus on syntax comparison. The model involves DNA translation and RNA transcription. For DNA translation, the DNA polymerase (DNA_Poly) binds to promoters (P2, Ptet) in a DNA strand and moves along the DNA to produce strands of messenger RNA (Mrna_P2, Mrna_Ptet). For RNA transcription, the ribosome (Ribo) binds to a messenger RNA and moves along it to produce proteins (GFP, RFP, CFP, GFP_F_RFP).

We start by defining the species:

```
from mobspy import *

Promoter, Start_Positions, Tet, Mortal = BaseSpecies()
Promoter.inactive, Promoter.active
Ribo, DNA_Poly = New(Start_Positions)
P2, Ptet = New(Promoter)
Mrna_P2, Mrna_Ptet, GFP, RFP, CFP, GFP_F_RFP = New(Mortal)
```

Further, we add death reactions:

```
# Death reactions
Mortal >> Zero [1]
```

In this model, promoters can either always be available for binding with the DNA polymerase (P2) or require binding with a BCRN species beforehand to bind to the DNA polymerase (Ptet). This is abstracted through the active and inactive characteristics:

```
# Promoter activation - only Ptet is activated. P2 is always
```

```
active Rev[Ptet.inactive + Tet >> Ptet.active][1, 1]
```

Translation and transcription are both implemented within a single function named `read`. DNA polymerase or ribosome meta-species are passed as the second argument of the function `R` for translation and transcription, respectively. The first parameter (`Pro`) is the binding target for `R`, being either a promoter located in the DNA strand for translation or a messenger RNA for transcription.

The product depends on the position where `R` initially binds on the DNA or RNA strand and where it stops reading. For instance, the protein `GFP_F_RFP` is generated when the ribosome begins at the `GFP` encoding position and continues into the `RFP` encoding without terminating, resulting in a fusion protein combining `GFP` and `RFP`. This is implemented inside the function using a `for` loop by passing the positions and respective products as the third argument `strand`.

```
# Read expresses the reading of RNA and DNA - both transcription
    and translation
# For translation, DNA polymerase is used; for mRNA, it
    is the Ribosome
    def read(Pro, R, strand):
  sp = 'started_' + strand[0][0] # sp stands for start position
  Start_Positions.c(sp)
  rate = [lambda r1, r2: 1 if r1.inactive else 2, 1]
  # From free state to bound to a site
  # for Ribosome it's free_Ribo,
    for DNA_poly it's free_DNA_Poly
  Rev[Pro + R.c('free_' + str(R)) >> Pro + R.c(sp).c('at_'
    + strand[0][0])] [rate] for (location, _), (next_l,
    Product) in zip(strand, strand[1:]):
    # Movement of the reader R.c(sp).c('at_' + location) >>
    R.c(sp).c('at_' + next_l) [1]
    # Remove reader from location and produce the resulting
        protein
    R.c(sp).c('at_' + next_l) >> R.c(sp).c('free_' + str(R))
     + Product[1]

# Zero produces nothing. If R unbinds at the starting
     position site, nothing is produced read(Ptet, RNA_Poly,
     [('ptet_dna', Mrna_Ptet), ('ptet_end', Zero)])
read(P2, RNA_Poly, [('p2_dna', Mrna_P2), ('p2_end', Zero)])
read(Mrna_Ptet, Ribo, [('gfp', GFP), ('rfp', GFP_F_RFP),
    ('rfp_end', Zero)])
read(Mrna_P2, Ribo, [('cfp', CFP), ('cfp_end', Zero)])
read(Mrna_Ptet, Ribo, [('rfp', RFP), ('rfp_end', Zero)])
```

Finally, we assign initial values to meta-species.

```
# P2 is set to active as it is a constitutive promoter
model = set_counts({RNA_Poly: 100, Ribo: 100, GFP: 0, RFP: 0,
     CFP: 0, Ptet: 1, P2.active: 1, Tet: 100, Mrna_Ptet: 0,
```

```
      Mrna_P2: 0, GFP_F_RFP: 0})
S = Simulation(model)
print(S.compile())
```

The resulting model has a similar length as the BioCRNpyler model. For MobsPy all the reactions are explicit in the model.

**Comparison to PySB.**  To compare MobsPy syntax to PySB syntax, we take the `hello_pysb.py` model from the PySB GitHub repository [36]:

```
from pysb import *

Model()

# Declare the monomers
Monomer('L', ['s'])
Monomer('R', ['s'])

# Declare the parameters
Parameter('L_0', 100)
Parameter('R_0', 200)
Parameter('kf', 1e-3)
Parameter('kr', 1e-3)

# Declare the initial conditions
Initial(L(s=None), L_0)
Initial(R(s=None), R_0)

# Declare the binding rule
Rule('L_binds_R', L(s=None) + R(s=None) | L(s=1) % R(s=1),
     kf, kr)
```

Implementing this model in MobsPy, we get:

```
from mobspy import *

L, R = BaseSpecies()

L_0, R_0, kf, kr = ModelParameters(100, 200, 1e-3, 1e-3)
L.sl_0 + R.sr_0 >> L.sl_1 + R.sr_1 [kf, lambda r: kr*r]

L(L_0), R(R_0)
S = Simulation(L | R)
print(S.compile())
```

Multiple differences in syntax choices are evident in this comparison. Firstly, the `BaseSpecies` constructor in MobsPy allows for the simultaneous definition of multiple meta-species, streamlining the process of declaring several monomers simultaneously. Additionally, characteristics that define bindings are implicitly added to reactions, eliminating the

need for users to manually define these characteristics when creating base species and typing them again in subsequent reactions.

PySB automatically generates Python variables using the string names provided in its `Monomer` constructor, blending string literals with variable references. While this approach works and compiles correctly, it can lead to namespace pollution. Namespace pollution occurs when dynamically injected variables create conflicts or unintended overwrites, especially in environments where multiple modules are imported or interact. It can result in hard-to-debug issues, as variable name conflicts may propagate across modules [37].

MobsPy achieves the same functionality by using the variable name itself as the internal string representation. This eliminates the need for the user to manage or interact with string representations directly. Instead, users work exclusively with variables. Also, users can import MobsPy's global variables with `import mobspy as ms` instead of `from mobspy import *` if namespace pollution is ever presented as an issue.

On the other hand, PySB has more synthesis capabilities due to its high-level modularity features. For MobsPy to achieve a similar level of syntax reduction, a necessary step is to implement ways to combine high-level models.

**Comparison to BioNetGen.** In this section, we compare MobsPy to BioNetGen. To differentiate with the comparison with Kappa, we chose a multi-state model available in VCell's introductory example list [24,38]:

```
begin parameters
 R0  100
 L0  500
 A0  100
 kon  0.01
 koff  0.1
 kAon  0.01
 kAoff  0.1
 kAp  0.01
 kAdp  0.1
end parameters

begin molecule types
 R(l,a)
 L(r)
 A(r,Y~U~P)
end molecule types

begin seed species
 R(l,a)  R0
 L(r)  L0
 A(r,Y~U)  A0
end seed species

begin reaction rules
 R(l) + L(r) <-> R(l!1).L(r!1) kon,koff
 R(a) + A(r) <-> R(a!1).A(r!1) kAon,kAoff
 L().R().A(Y~U) -> L().R().A(Y~P)  kAp
 A(Y~P) -> A(Y~U)  kAdp
```

```
end reaction rules

begin observables
 Molecules A_P   A(Y~P)
 Molecules A_unbound_P   A(r,Y~P)
 Molecules A_bound_P   A(r!+,Y~P)
 Molecules RLA_P   R().L().A(Y~P)
end observables

generate_network();
writeSBML();
simulate_ode({t_end=>50,n_steps=>20});
```

Translated into MobsPy, this model becomes:

```
R0, L0, A0, kon, koff, kAon, kAoff, kAp, kAdp =
    ModelParameters(100, 500, 100, 0.01, 0.1, 0.01, 0.1,
        0.01, 0.1)

r_link, C = BaseSpecies()
r_link.r_0, r_link.r_1
R, L, A = New(r_link)

R.r_0 + L.r_0 >> R.r_1 + L.r_1 [kon, lambda r: koff*r]

R.a_0 + A.r_0 >> R.a_1 + A.r_1 [kAon, lambda r: kAoff*r]

C.assign(A.r_1.n_p - R.a_1.r_0)
C + A.r_1.n_p >> C + A.r_1.y_p [lambda C: kAp*C]

A.y_p >> A.n_p [kAdp]

R(R0), L(L0), A(A0)
S = Simulation(R, L, A)
print(S.compile())
```

Some syntax advantages gained from inheritance can be seen in the definition of R, L, A. Since all three inherit from r_link, there is no need to repeatedly redefine binding states when creating the molecules. Furthermore, MobsPy implicit state definition is used to define:

- Phosphorylation of A is represented by y_p (phosphorylated) and n_p (unphosphorylated).
- R binding to A is a_0 (unbound) and a_1 (bound).

Finally, observables do not need to be explicitly defined in MobsPy. Instead, the user can query the simulation results directly from the generated data.

In the third meta-reaction, which is complex-based, BioNetGen uses its graph rewriting algorithm to identify and count complexes, defining the reaction rate accordingly. In MobsPy,

we achieve the same effect by explicitly defining a complex species (C) assigned the number of receptor-ligand complexes containing unphosphorylated A. The reaction rate is based on C, ensuring that only A.n_p inside a complex contributes to the phosphorylation reaction, rather than all A.n_p in the system. This expression is simple to write for small complexes. However, as the complex grows in size, so does the expression in the assignment.

It is vital to note that this model considers the binding rate of molecules linked to other parts of the complex equal to the binding rate of the molecules when they are not bound. One example lies in the dephosphorylation of A, which is of equal rate when A is in a complex or not. It qualifies as an approximation, as in most chemical systems, the binding of other complexes and the size of the complexes affect the reaction rate constant both due to electromagnetic interactions between adjacent bond sites and changes in overall complex mass [39–41].

**Comparison to antimony.** Unlike the other languages presented in this section, Antimony does not provide reaction modularity. Instead, it serves as a human-readable alternative to SBML, offering additional features such as modifying the CRN structure mid-simulation[18]. Since MobsPy supports Antimony model generation through the `generate_antimony` method, we employ this feature for comparison.

The following MobsPy script generates an Antimony model:

```
from mobspy import *

Replicator, Mortal = BaseSpecies()
Replicator >> 2*Replicator [lambda r: 2*(100 - r)*r]
Mortal >> Zero [1]

A, B, C = New(Replicator*Mortal)
A + B >> C [1]

S = Simulation(A | B | C)
print(S.generate_antimony()[0])
```

This script produces the following Antimony model:

```
model mobspy_25681
 A = 0 dimensionless
 B = 0 dimensionless
 C = 0 dimensionless
 _vol = 1 dimensionless
 reaction_0: A -> ; A * 1
 reaction_1: B -> ; B * 1
 reaction_2: C -> ; C * 1
 reaction_3: A + B -> C; A * B * 1 * _vol^-1
 reaction_4: A -> 2 A; ((2*(100-A))*A)
 reaction_5: B -> 2 B; ((2*(100-B))*B)
 reaction_6: C -> 2 C; ((2*(100-C))*C)
end
```

This output demonstrates how MobsPy expressions integrate with Antimony, as both directly encode reaction rates using mathematical expressions.

## Availability and future directions

MobsPy is freely available at the Python Package Index (via pip) and its GitHub repository:

https://github.com/ROBACON/mobspy

Future updates will focus on enhancing modularity by developing a library of reusable meta-species models, particularly for translation and transcription. Additionally, we aim to refine the syntax to improve integration with existing models. Finally, we plan to enhance the current geometry models by introducing methods that simplify mesh definition while mitigating state space explosion. Finally, we aim to implement a syntax that automates complex-based assignments.

## Supporting information

**S1 Appendix. Implementation details and plot configurations.**
(PDF)

## Author contributions

**Conceptualization:** Fabricio Cravo, Matthias Függer, Thomas Nowak.

**Formal analysis:** Fabricio Cravo, Matthias Függer, Thomas Nowak.

**Funding acquisition:** Matthias Függer, Thomas Nowak.

**Investigation:** Fabricio Cravo, Gayathri Prakash, Matthias Függer, Thomas Nowak.

**Methodology:** Fabricio Cravo, Matthias Függer, Thomas Nowak.

**Project administration:** Matthias Függer, Thomas Nowak.

**Resources:** Matthias Függer, Thomas Nowak.

**Software:** Fabricio Cravo, Gayathri Prakash, Matthias Függer, Thomas Nowak.

**Supervision:** Matthias Függer, Thomas Nowak.

**Validation:** Fabricio Cravo, Matthias Függer, Thomas Nowak.

**Visualization:** Fabricio Cravo, Gayathri Prakash, Matthias Függer, Thomas Nowak.

**Writing – original draft:** Fabricio Cravo, Matthias Függer, Thomas Nowak.

**Writing – review & editing:** Fabricio Cravo, Gayathri Prakash, Matthias Függer, Thomas Nowak.

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
