## [Decision Letter · Decision Letter 0]

19 Dec 2024

PCOMPBIOL-D-24-01024

MobsPy: A Programming Language for Biochemical Reaction Networks

PLOS Computational Biology

Dear Dr. Nowak,

first, we would like to apologize for the long wait. 

We were hoping to receive additional input from a potentially important reviewer. But we don't want to let you wait any longer. 

Based on the reviewer comments, we would like to invite you to submit a revised version of the manuscript that addresses the points raised during the review process. Unfortunately, we cannot promise acceptance of a revised manuscript at this point.  

Please submit your revised manuscript within 60 days Feb 18 2025 11:59PM. If you will need more time than this to complete your revisions, please reply to this message or contact the journal office at ploscompbiol@plos.org. Please include the following items when submitting your revised manuscript:

We look forward to receiving your revised manuscript.

Kind regards,

Martin Meier-Schellersheim

Academic Editor

PLOS Computational Biology

Stacey Finley

Section Editor

PLOS Computational Biology

**Journal Requirements:**

2) Your manuscript is missing the following sections: Design and Implementation, and Availability and Future Directions. Please ensure that your article adheres to the standard Software article layout and order of Abstract, Introduction, Design and Implementation, Results, and Availability and Future Directions. For details on what each section should contain, see our Software article guidelines:

https://journals.plos.org/ploscompbiol/s/submission-guidelines#loc-software-submissions

4) We notice that your supplementary Figures are included in the manuscript file. Please remove them and upload them with the file type 'Supporting Information'. Please ensure that each Supporting Information file has a legend listed in the manuscript after the references list.

Potential Copyright Issues:

- Figure 1A; Please confirm whether you drew the images / clip-art within the figure panels by hand. If you did not draw the images, please provide a link to the source of the images or icons and their license / terms of use; or written permission from the copyright holder to publish the images or icons under our CC BY 4.0 license. Alternatively, you may replace the images with open source alternatives. See these open source resources you may use to replace images / clip-art:

**Reviewers' comments:**

Reviewer's Responses to Questions

**Comments to the Authors:**

Reviewer #1: The article presents MobsPy, a Python-based language designed to simplify the modeling of Biochemical Reaction Networks (BCRNs). The authors introduce the concept of meta-species and meta-reactions to manage the complexity of BCRNs, enabling arguably more concise and manageable models. The paper highlights the challenges of modeling complex biochemical systems and proposes MobsPy as a tool to reduce errors and simplify the creation and maintenance of models. While I agree that the creation of models using species characteristics will be much easier using MobsPy, I’m not sure that parameterization, annotation and maintenance will also be simplified.

The article provides a thorough comparison with other frameworks, effectively highlighting MobsPy’s advantages. Though I found it cumbersome to have to look up the original examples from their respective websites. One aspect I miss in the translated MobsPy version are the actual parameter names used to make it comparable (so for example:

A >> 2*B [mobspy_parameters.Mobspy_Parameter('k1', 3)]

B >> C + D [mobspy_parameters.Mobspy_Parameter('k2', 1.4)]).

It would have been nice if the authors had chosen a direct comparison with PySB as well.

The software itself is easily installable and the github repository contains the examples and works as advertised.

The main questions left for me are:

• The MobsPy language uses ‘>>’ to denote an irreversible transition. Is there a way to define a reversible one?

• It would be nice if in the compiled model, there would be a way to easily use parameters instead of the number literals. This becomes especially important later for parameter estimation or entering values from literature for specific reactions. Is there a shortcut for that?

• It would be easier to use the software if there was a detailed readthedocs site with the examples and api description. I see the authors started with one, it would be great to see it succeed.

• It would also be nice if the examples / notebooks covered the SBML export explicitly. Especially together with using non-default rate laws.

• Looking at the SBML generated, I see a need for improvement:

o The exported sbml contains a dimensionless volume parameter, that is used instead of SBML compartments (though the exported compartment ‘c1’ is also dimensionless). This will make multi-compartment modelling difficult and error prone later, when the authors work on model composition.

o In fact, the whole SBML generated is unitless since dimensionless units are also used for species. It would be nice if the units (which the authors recognize as important since they added support for them for the mobspy language through pint) could also be exported to the SBML.

• MobsPy also exports Antimony script, however, the generated script is unfortunately also invalid. The reaction scheme generated is invalid. For example reaction schemes like ‘A B -> 2*C D;..’ are created, when it should be ‘A + B -> 2 C + D;…’

• Consistent spelling should be used for the main concepts like meta-species and it should be consistently hyphenated.

• For roadrunner and basico the publications for the tools should be used rather than their website.

Reviewer #2: Manuscript describes an agent-based approach for modeling, where the authors introduced multi-state agents that are termed by authors as “meta-species”, and interactions between meta-species that generate the reaction network to be simulated stochastically.

The approach is very interesting and should be useful for many systems biology modelers. However, the present manuscript suffers from a major drawback – the authors don’t compare their approach to the existing approaches and tools that aim at the same target - designing a language to simplify the modeling processes. Because of that, the manuscript has multiple overstatements, missing citations, or misrepresents other software tools.

1. Regarding human-readable description, the Antimony serves the same purpose. A significant part of the manuscript talks about assignments, events, and other elements of SBML modeling. It would be useful to compare with Antimony (providing the same model coded in both languages) and state the advantages of Antimony (e.g. better coverage of SBML) and mobspy (elements of rule-based modeling).

2. Regarding the introduction of meta-species, it’s a reinvented name of “multi-state molecule” that was introduced first by STOCHSIM back in 2001 (doi:10.1093/bioinformatics/17.6.575). The first general-purpose software that used multi-state molecules was BioNetGen software (doi:10.1093/bioinformatics/bth378), which seems to be roughly equivalent to the MobsPy declaration of meta-species. The BioNetGen in 2004 expanded the rules of interactions between multi-state species into the reaction network. The tricks with link1 and link2 were possible, but any complexes of several connected molecules were difficult to model, leading to the need to introduce bonds in addition to states, as described in doi: 10.1007/978-1-59745-525-1_5.

3. The authors should compare rule-based features of MobsPy to those of BioNetGen and Kappa. Modeling of complex formation with MobsPy is much trickier. Including several constraints into the interactions (A can bind to C if A is bound to B and C has a phosphorylation state and is in a complex with other molecules) is tricky if possible. Modeling of polymers is impossible. However, MobsPy excels in modeling just multi-state molecules, without binding. For example, the reaction A.state1 -> A.state1 + A is easy in MobsPy. However, both BioNetGen and Kappa can only match one reactant to one product, so the second A will give an error if not defined, which requires multiple reaction rules (r3-r3_4) to define all combinations. See below the BioNetGen model that implements the tree model:

begin model

begin molecule types

Tree(age~young~old,color~green~yellow~brown,location~dense~sparse)

Dead_tree()

end molecule types

begin seed species

Tree(age~young,color~green,location~dense) 25.0

Tree(age~old,color~green,location~dense) 25.0

Tree(age~young,color~green,location~sparse) 25.0

Dead_tree() 0.0

Tree(age~old,color~green,location~sparse) 25.0

end seed species

begin observables

Molecules All_Trees Tree()

Molecules Young_trees Tree(age~young)

Molecules Brown_trees Tree(color~brown)

Molecules Dense_trees Tree(location~dense)

Molecules Sparse_trees Tree(location~sparse)

end observables

begin reaction rules

r00: Tree(age~young) -> Tree(age~old) 0.1

r01: Tree(age~old) -> Dead_tree() 0.1

r02: Tree(age~old,location~dense)%1 + Tree(age~young,location~dense) -> Tree(age~old,location~dense)%1 1.0E-10

r03: Tree(age~old,color~green,location~dense)%1 -> Tree(age~old,color~green,location~dense)%1 + Tree(age~young,color~green,location~dense) 0.1

r03a: Tree(age~old,color~green,location~sparse)%1 -> Tree(age~old,color~green,location~sparse)%1 + Tree(age~young,color~green,location~sparse) 0.1

r03b: Tree(age~old,color~yellow,location~dense)%1 -> Tree(age~old,color~yellow,location~dense)%1 + Tree(age~young,color~yellow,location~dense) 0.1

r03c: Tree(age~old,color~yellow,location~sparse)%1 -> Tree(age~old,color~yellow,location~sparse)%1 + Tree(age~young,color~yellow,location~sparse) 0.1

r03d: Tree(age~old,color~brown,location~dense)%1 -> Tree(age~old,color~brown,location~dense)%1 + Tree(age~young,color~brown,location~dense) 0.1

r03e: Tree(age~old,color~brown,location~sparse)%1 -> Tree(age~old,color~brown,location~sparse)%1 + Tree(age~young,color~brown,location~sparse) 0.1

r04: Tree(color~green) -> Tree(color~yellow) 10.0

r05: Tree(color~yellow) -> Tree(color~brown) 10.0

r06: Tree(color~brown) -> Tree(color~green) 10.0

end reaction rules

end model

4. Implementation and use of MobsPy should be compared to PySB – whether the same model can be modeled in PySB and MobsPy, and which is easier to use.

5. I recommend comparing MobsPy to BIOCHAM (http://contraintes.inria.fr/biocham) which works with multi-state molecules. I also recommend looking at ML-rules (doi: 10.1007/978-1-4939-9102-0_6)

6. Some statements in the manuscript are incorrect:

a. Page3: “Common to the above tools, however, is that they lack means for abstraction at the reaction level”. What do authors mean by “abstraction”? All these tools provide visualization of the reaction network, much better than MobsPy code. If rule-based features are counted as “abstraction”, then there are other tools that have them, e.g. SmolDyn (doi: 10.1093/bioinformatics/btw700) and VCell (doi:10.1093/bioinformatics/btw353) that implement BioNetGen muti-state rule-based modeling. CellDesigner implements some multi-state features.

b. Page 3: “Unlike rule-based languages, MobsPy’s meta-species characteristics can be virtually anything assigned by the user, accommodating a wide range of possibilities according to intent” – the same is true for rule-based approach, the term “molecule” should not confuse – it’s just an object that can be anything.

c. Page4: Parameter sweeps, utilization of multiple CPUs are the features of Python and BasiCO, not MobsPy. PySB can do the same.

d. Page 15, description of Kappa, BioNetGen, and pySB: “the characteristics of a species, or, equivalently, its state, are limited to the occupancy status of their binding sites” – wrong, there are states that are not limited to occupancy, see the BioNetGen example I demonstrated above.

e. Page 16: “PySB, although lacking built-in simulation capabilities” – the same is true for MobsPy which lacks simulation capabilities and uses external simulators.

f. Page 16: “Current tools that provide reaction modularity, like Kappa and BioCRNPyler [27, 10], do not use built-in units” – wrong, rule-based modeling with VCell uses built-in units. The Tree model in the manuscript provides a plot in counts while GitHub example results in simulation results in concentrations – ironically as the authors talk about how great their tool is with units.

7. The models described in the manuscript don’t match the simulation plots, and do not match the tutorials at GitHub. I started to deal with the first “tree” example and I tried to reproduce it in BioNetGen just from the manuscript. It was impossible: initial conditions start from 75, while plots show 100. My simulations have at 20 years total trees down to 20%, while they didn’t change in the manuscript plots. Then I loaded tutorial number 1, and got simulation results matching my BioNetGen simulations but not in the manuscript. Next, I loaded a model from Examples folder – and got simulation results matching “qualitatively” plots in the manuscript but not numbers in the code snippets in the manuscript. I had to go line by line to figure out the difference in numbers (the death rate is 10 times higher in the tutorial than in the manuscript text and the first tutorial notebook).

8. Same with all other examples. The reference [22] does not have the plots given in this manuscript – how are readers supposed to believe that the simulations are correct?

As conclusion, I would appreciate:

- Detailed comparison of MobsPy to other tools that have multi-state features, listing pros and cons. Some of the tools that use rule-based technology are listed at https://bionetgen.org/software

- Choose several tools (most obvious PySB, BioNetGen and/or Kappa) and demonstrate models/features that can be simulated by all tools (including simulations and plots), and features that are unique to MobsPy. You can use https://bionetgen.org/applications and https://bnglviz.github.io/examples.html as a collection of examples that are covered by BioNetGen modeling.

- All code snippets in the manuscript should match precisely those in the examples folder

- All plots in the manuscript should be precisely reproducible from the examples in the GitHub.

- Names of models in the examples folder should be given in the manuscript. Now it’s a quest to find the matching model.

- If a model claims to reproduce the published results, please point out to the exact figure in the original manuscript you reproduce.

Minor issues:

1. Colors are completely decoupled from the rest of the Tree model – every modeler would just create a separate model for color transition. A simple model where colors will depend on other features like age will be much more interesting.

2. I got strange warning messages when running notebook 1:

Compiling model

WARNING: The following reaction:

{'re': [(1, 'Tree_dot_brown_dot_dense_dot_old')], 'pr': [(1, 'Tree_dot_brown_dot_dense_dot_old'), (1, 'Tree_dot_brown_dot_dense_dot_young')], 'kin': 'Tree_dot_brown_dot_dense_dot_old * 3.168808781402895e-09'}

Is doubled. Was that intentional?

3. Please delete numbers in the code snippets in the manuscript – it makes copying to Python complicated.

**Have the authors made all data and (if applicable) computational code underlying the findings in their manuscript fully available?**

Reviewer #1: Yes

Reviewer #2: Yes

PLOS authors have the option to publish the peer review history of their article (what does this mean?). If published, this will include your full peer review and any attached files.

Reviewer #1: No

Reviewer #2: No

**Figure resubmission:**
---

## [Decision Letter · Decision Letter 1]

4 Apr 2025

Dear Nowak,

We are pleased to inform you that your manuscript 'MobsPy: A Programming Language for Biochemical Reaction Networks' has been provisionally accepted for publication in PLOS Computational Biology.

Best regards,

Martin Meier-Schellersheim

Academic Editor

PLOS Computational Biology

Stacey Finley

Section Editor

PLOS Computational Biology

Reviewer's Responses to Questions

**Comments to the Authors:**

Reviewer #1: Thank you for addressing all my previous concerns. The new read the docs page, additional comparisons, and corrected antimony generation is much appreciated.

I hope in a future version of the tool you will add the export of pint units to the generated antimony / sbml. As looking at the export in (https://mobspy-doc.readthedocs.io/en/latest/example_models/tutorial_notebooks/17_Export_Model.html#id1) I find the numeric exported values confusing. And personally, I would prefer the resulting model not being dimensionless.

When using the Rev operator (same example linked above), it would be great if the resulting antimony and SBML would also have the reactions marked as reversible which is not the case right now.

minor:

- some typos in the manuscript that should be fixed e.g. 'assigments, nuymber', consistent hyphenation eg. object oriented vs object-oriented, duplication eg. 'that act that act'

- it would be great if COPASI could be spelled in all caps,

Reviewer #2: The authors fully addressed all comments.

**Have the authors made all data and (if applicable) computational code underlying the findings in their manuscript fully available?**

Reviewer #1: Yes

Reviewer #2: Yes

PLOS authors have the option to publish the peer review history of their article (what does this mean?). If published, this will include your full peer review and any attached files.

Reviewer #1: No

Reviewer #2: No

---

## [Editor Report · Acceptance letter]

PCOMPBIOL-D-24-01024R1

MobsPy: A Programming Language for Biochemical Reaction Networks

Dear Dr Nowak,

I am pleased to inform you that your manuscript has been formally accepted for publication in PLOS Computational Biology. Your manuscript is now with our production department and you will be notified of the publication date in due course.

With kind regards,

Anita Estes
